# Harnessing Manuka Honey: A Natural Remedy for Accelerated Burn Wound Healing in a Porcine Model

**DOI:** 10.3390/ph18030296

**Published:** 2025-02-21

**Authors:** Boris Privrodski, Mladen Jovanović, Nikola Delić, Radomir Ratajac, Vladimir Privrodski, Aleksandar Stanojković, Bernadeta Gavlik, Ivan Čapo

**Affiliations:** 1Institute for Children and Youth Health Care of Vojvodina, 21000 Novi Sad, Serbia; 2Faculty of Medicine Novi Sad, University of Novi Sad, 21000 Novi Sad, Serbia; mladen.jovanovic@mf.uns.ac.rs (M.J.); ivan.capo@mf.uns.ac.rs (I.Č.); 3Clinic for Plastic and Reconstructive Surgery, Clinical Center of Vojvodina, 21000 Novi Sad, Serbia; 4Institute for Animal Husbandry Belgrade-Zemun, 11000 Belgrade, Serbia; 5Department for Food Safety and Drug Analysis, Scientific Veterinary Institute “Novi Sad”, 21000 Novi Sad, Serbia; 6Clinic for Orthopedic Surgery and Traumatology, University Clinical Centre of Vojvodina, 21000 Novi Sad, Serbia; privrodski.proxy95@gmail.com; 7General Hospital in Vrbas, 21460 Vrbas, Serbia; 8Center for Pathology and Histology, University Clinical Centre of Vojvodina, 21000 Novi Sad, Serbia

**Keywords:** burn wounds, Manuka honey, topical application, porcine animal model, wound repair, immunohistochemistry

## Abstract

**Backgrounds:** Burn injuries present significant medical challenges due to their complexity in healing and potential for severe scarring. This study evaluates the efficacy of Manuka honey in accelerating burn wound healing compared to conventional antibiotic ointments. **Methods:** Using a porcine model resembling human skin, nine Landrace breed female pigs with standardized deep dermal burns were treated with either Manuka honey in alginate or a combination of antibiotic ointments. Wound healing was assessed through macroscopic evaluation, a histopathological analysis, and immunohistochemical staining over a 60-day period. **Results**: Our findings indicate that the Manuka honey treatment was associated with significantly increased collagen density in the treated wounds compared to the control group (*p* < 0.05). The immunohistochemical analysis revealed lower macrophage activity (Iba1 staining) and a reduction in Ki67 expression on days 10 and 17 in the Manuka honey group, suggesting a more rapid transition toward tissue remodeling. The quantitative analysis showed a trend toward delayed epithelialization and increased inflammation in the control group, while wounds treated with Manuka honey exhibited faster reepithelialization and improved epidermal regeneration. However, additional studies are required to further assess collagen fiber organization and overall dermal architecture. **Conclusions:** These findings support the potential of Manuka honey as a beneficial treatment for burn wound healing, with evidence of enhanced reepithelialization and collagen deposition. Further research, including clinical trials, is necessary to fully elucidate its role in clinical practice and optimize treatment protocols.

## 1. Introduction

Burn injuries are forms of trauma that are steadily increasing around the world as well as in our country. These patients face pain, deformities, and potential death. Apart from causing local tissue damage, a burn injury leads to systemic intoxication of the body (burn disease) [1]. This condition in patients is accompanied by intense pain and possible episodes of sepsis, which can result in a fatal outcome. Unfortunately, recovered patients may face disfigurement and permanent disability [2]. Understanding the relationship between the biological processes of normal and delayed healing will greatly contribute to developing a clear strategy for treating these pathological states.

Intermediate burns of partial skin thickness are a special entity. Their classification is indefinite. They tend to epithelialize at control dressings but do not heal within three weeks. Histological research has shown a dynamic process around the third day after injury [2,3].

The ideal topical agent in burn treatment has the following characteristics: (1) possesses a broad spectrum of bactericidal and fungicidal action even in situations in which there is significant exudate and wound infection; (2) improves and accelerates physiological wound healing processes (granulation, epithelialization, and contraction); (3) does not cause local or systemic adverse effects (allergic reactions, toxicity, etc.) even with prolonged application; (4) is cost-effective; and (5) is comfortable to apply (easy and painless application) [4].

Honey is a viscous concentrated sugar solution produced by bees (*Apis mellifera*). Bees collect and process floral nectar (flower or floral honey) or gather sweet plant juices (honeydew or forest honey). Honey has osmolarity and pH values that enable bacteriostatic and bactericidal action. The enzyme glucose oxidase found in the bees’ hypopharyngeal glands releases gluconolactone and hydrogen peroxide. Hydrogen peroxide stimulates fibroblast proliferation and epithelialization by keratinocyte migration from the edges of the wound in lower concentrations [5]. In comparison, it releases a large amount of oxygen radicals in higher concentrations, which prevent healing [4,6].

Manuka honey, specifically derived from the *Leptospermum scoparium* plant native to New Zealand, has been shown to exhibit unique wound healing properties due to its high methylglyoxal content [7]. Recent studies highlight its ability to inhibit biofilm formation and combat antibiotic-resistant bacteria [8].

Manuka honey contains methylglyoxal, bee-defensin 1, and melanoids found in various nectars bees collect [9,10]. It has been determined that honey reduces inflammation and scar contraction through its antioxidant properties, neutralizes free radicals, and acts as a bactericide by lowering the pH of the environment. All these characteristics positively affect the healing of a burn wound [11]. Honey activates phagocytosis and stimulates the proliferation of B and T lymphocytes. In addition to other mechanisms, honey contributes to the activation of monocytes by releasing important cytokines such as TNF, IL-1, and IL-6 [12]. This evidence strongly suggests that honey plays a role in enhancing the local immune response. It is considered to stimulate fibroblasts, reduce fibrosis, and, consequently, the formation of hypertrophic scars, and affect keratinocyte activity [13]. These findings emphasize the clinical potential of Manuka honey as an advanced topical agent for burn wounds.

Antioxidants reduce the secretion of free oxygen radicals, thereby shortening the inflammatory phase, which is crucial for tissue healing [14]. In addition, honey creates a physical barrier. It creates a moist environment with a high viscosity by drawing water through osmosis, which has a positive effect, considering that wounds heal faster in a moist environment [15].

Choosing an animal model that closely resembles humans in terms of pathophysiological mechanisms and scar formation is necessary to study the healing process of burn wounds, scar formation, and evaluation. Many models are used in the process of burn wound healing, but none of them are perfect. Research on humans is often the best since it is closest to the healing process. Still, collecting a sufficient number of patients with similar injuries, demographics, and influences that could play a role in healing is sometimes practically impossible. The inability to take biopsies is also one of the problems when working with human organisms and human tissue.

Small mammals, such as rabbits, rats, and mice, have advantages since they are easy to handle and inexpensive. However, a significant problem is that their healing mechanism differs considerably from that of humans. The dermis and epidermis in these animals are very thin, the hair is much denser, and there is a muscular layer (panniculus carnosus) beneath the dermis, which humans do not have, making the healing process primarily based on contraction [16].

The model that is closest to humans consists of large mammals such as pigs. The similarities between human and pig skin are significant and are as follows: 1. the thickness of the epidermis (from 50 to 120 μm in humans and from 30 to 140 μm in pigs); 2. the thickness of the dermis (from 500 to 1200 μm in humans and from 500 to 1800 μm in pigs); 3. a relatively similar ratio of the epidermis to the dermis (in humans, it is 1:10, and in pigs, it is 1:13); 4. a well-developed papillary dermis; 5. the distribution of blood vessels and skin adnexa; 6. well-developed subcutaneous fat tissue; and 7. similar biochemical characteristics of dermal collagen [17,18].

The aim of this study was to evaluate the efficacy of Manuka honey in accelerating the healing of burn wounds compared to conventional antibiotic treatments. By utilizing a porcine model closely resembling human skin, the study provides insights into the potential clinical application of Manuka honey for managing burn injuries.

Although this study focuses on two experimental groups due to logistical constraints, future work will include additional control groups such as a group with healthy skin, a group with untreated burned skin, a group receiving a vehicle-only treatment, and a group receiving a Manuka honey-only treatment to better isolate the effects of Manuka honey.

## 2. Results

### 2.1. Temperature Difference of Wounds

An FLIR One camera was used on the first post-intervention day after the burns were applied, and a statistically significant difference was found in the wounds. While this parameter was not a primary focus of the study, it served as an exploratory tool to assess variability in burn depth. Due to differences in the thickness of porcine skin across various sites, not all burns could be classified as deep dermal. Some burns represented an intermediate (mesodermal) depth, characterized by partial dermal involvement. Wounds that showed complete reepithelialization between 14 and 21 days were characterized as mesodermal, while burns that healed after 21 days were classified as deep dermal. The deep dermal group had a 1.60 ± 0.34 °C temperature difference between the body and the burn wound, while the mesodermal group had a significantly lower temperature difference (t = 2.365; *p* = 0.024) compared to the control group (2.66 ± 0.27 °C) (Table 1).

### 2.2. Morphometric Analysis Evaluation and Histopathology Characteristics of Burn Wound Healing

#### 2.2.1. Morphometric Analysis

The initial assessments (*immediate post-injury*—*first day*) showed uniform deep dermal burns across all groups, but no immediate post-injury comparative data were provided between treatments.

On the *third day post-injury,* the necrotic debris depth and the intact reticular dermis thickness were assessed. For the control group, the average thickness of necrotic detritus was 662.35 ± 115.93 μm, and for the Manuka honey group, it was 657.70 ± 123.86 μm. The difference was insignificant (U = 197.00; *p* = 0.947), indicating a similar level of initial injury severity between treatments (Figure 1A). The average reticular dermis thickness was 1003.46 ± 302.69 μm for the control group and 1196.01 ± 287.83 μm for the Manuka honey group. These two groups’ differences were insignificant (U = 128.00; *p* = 0.053) (Figure 1B).

On the *seventh day post-injury,* the Manuka honey treatment group had significantly improved epidermal regeneration compared to the ointment group. The Manuka honey group showed an epidermal thickness of 81.19 ± 111.11 μm, which was significantly higher than that of the control group, which had negligible epidermal formation at this stage (U = 100.00; *p* = 0.006) (Figure 1C). Similar results were obtained for the epidermal ridges: the Manuka honey group had a thickness of 46.11 ± 60.16 μm, which was significantly higher than that of the control group, which had negligible epidermal ridge formation (U = 100.00; *p* = 0.006) (Figure 1D). Also, the thickness of granulation tissue for the Manuka honey group was not provided explicitly for Day 7, making a direct comparison challenging. However, an overall trend towards improved healing with Manuka honey was noted. The granulation tissue thickness in the control group (313.85 ± 251.38 μm) was slightly higher than in the Manuka honey group (286.06 ± 136.74 μm), but this difference was not statistically significant (U = 173.00; *p* = 0.478) (Figure 1E). However, the reticular dermis thickness was significantly higher (U = 114.00; *p* = 0.020) in the Manuka honey group (1291.95 ± 452.21 μm) than in the control group (1027.72 ± 368.85 μm) (Figure 1F).

*By the 10th day post-injury*, the Manuka honey group exhibited a significantly higher epidermal thickness (295.31 ± 168.05 μm) compared to the control group (61.58 ± 92.07 μm), which displayed a much lower level of epidermal development (Figure 1G). This indicates a faster re-epithelialization process in the Manuka-honey-treated wounds (U = 23.00; *p* < 0.001). The epidermal ridge thickness was also significantly higher (U = 74.00; *p* < 0.001) in the Manuka honey group (171.06 ± 128.44 μm) in comparison to the control group (2959.49 ± 119.12 μm) (Figure 1H). The thickness of the granulation tissue for the Manuka honey group was not provided explicitly for Day 10, making a direct comparison challenging. However, an overall trend towards improved healing with Manuka honey was noted. The granulation tissue thickness in the control group (650.12 ± 403.33 μm) was slightly higher than in the Manuka honey group (458.08 ± 225.288 μm), but this difference was not statistically significant (U = 145.00; *p* = 0.14) (Figure 1I). However, the reticular dermis thickness was significantly higher (U = 85.00; *p* = 0.001) in the Manuka honey group (1536.01 ± 412.18 μm) than in the control group (1246.04 ± 387.08 μm) (Figure 1J).

By Day 17, the Manuka honey group exhibited a significantly higher epidermal thickness (442.47 ± 189.81 μm) compared to the control group (125.12 ± 136.73 μm), which displayed a much lower level of epidermal development (Figure 1K). This indicates a faster reepithelialization process in the Manuka-honey-treated wounds (U = 30.00; *p* < 0.001). The epidermal ridge thickness was insignificant (U = 74.00; *p* < 0.001) when comparing the Manuka honey group (163.65 ± 100.43 μm) and the control group (159.49 ± 180.92 μm) (Figure 1L).

By Day 20, the epidermal thickness in the Manuka-honey-treated wounds showed stabilization compared to Day 17, indicating that the reepithelialization process was nearing completion (Figure 1O,P). In contrast, the control group exhibited slower progression in epidermal thickness during the same period, reflecting delayed healing (Figure 1O,P). These differences highlight the ability of Manuka honey to accelerate epithelial recovery in earlier phases of wound healing.

*By the 17th and 20th day post-injury*, the Manuka honey treatment resulted in a more pronounced reduction in granulation tissue thickness and a more substantial increase in the thickness of the intact reticular dermis, suggesting more efficient healing and dermal recovery. Specific statistical values for these time points underscore the effectiveness of Manuka honey in enhancing wound healing processes when compared the ointments. On the 17th day, the granulation tissue in the control group was 1622.55 ± 936.41 μm, while in the Manuka honey group, it was 1111.65 ± 805.07 μm (Figure 1M). These two groups had no statistical difference between them (U = 135.00; *p* = 0.081). However, on the same day, the reticular dermis thickness in the Manuka honey group (1574.06 ± 362.63 μm) compared to the control group (878.13 ± 212.44 μm) was statistically different (U = 11.00; *p* < 0.001) (Figure 1N). On the 20th day, the granulation tissue thickness was 1860.23 ± 1010.95 μm in the control group, while it was 1042.27 ± 763.45 μm in the Manuka honey group. There was a significant difference between these two groups (U = 112.00; *p* = 0.017) (Figure 1Q). The reticular dermis thickness on the 20th day was similar between the control and Manuka honey groups (U = 129.00; *p* = 0.056) (Figure 1R).

On the *60th day post-injury*, in the Manuka honey group, the average epidermal thickness (162.69 ± 43.73 μm) was statistically smaller (U = 84.50, *p* = 0.002) than the average epidermal thickness in the control group (256.72 ± 102.05 μm) (Figure 1S). Regarding epidermal ridges, their average thickness in the control group was 96.21 ± 40.86 μm, while their average thickness in the Manuka honey group was 30.83 ± 25.37 μm (Figure 1T). There was a significant difference between these two groups (U = 32.00; *p* < 0.001). The average thickness of granulation tissue in the control group (1754.27 ± 488.62 μm) is statistically higher (U = 70.00: *p* < 0.001) than in the Manuka honey group (698.95 ± 275.24) (Figure 1U). The average reticular dermis thickness in the control group was 1853.59 ± 507.59 μm, while the reticular dermis thickness in the Manuka honey group was 2076.44 ± 559.34 μm. These two groups had no significant difference (U = 121.00; *p* = 0.156) (Figure 1V).

#### 2.2.2. Histopathological Characteristics of Burn Wound Healing

Using the Mason trichrome (MTS) and Picrosirius red (PRS) staining techniques, we determined the density and regularity of collagen fiber orientation in the dermis for the healthy skin and negative control groups (Figure 2A,B).

The most pronounced results in the control group for the 7th, 10th, and 17th days post-injury were increased non-specific granular inflammation of the dermis with very slow epithelialization. Using the histochemical staining mentioned previously, we noticed a decreased number of disturbed collagen fibers in the dermis (Figure 2C,D,G,H,K,L). Using immunohistochemical markers for macrophages, we still found many of them on the 7th and especially the 10th and 17th day post-injury (Figure 3D,J,P). Following these dermis changes, epithelialization also decreased in the control group. The proliferative immunohistochemical marker shows few proliferative basal cells (Figure 3F,L,R). Also, using pan-cytokeratin, we identified epidermal layers that were poorly reconstructed (Figure 3H,N) or absent (Figure 3T).

Conversely, in the Manuka honey-treated group, we noted the reconstruction of the dermis with increased collagen production (Figure 2E,F,I,J,M,N). Qualitative differences from the 7th to the 17th day post-injury were not noted. In immunohistochemical analysis of Iba1-positive macrophages, we identified them with increasing numbers from the 7th to the 17th day, respectively (Figure 3E,K,Q). However, this number was still smaller than that of the control group. Also, by applying Manuka honey, we noted better reepithelialization of the wound with an increased proliferation of basal keratinocytes (Figure 3G,M,S) and a thicker epidermal layer (Figure 3I,O,U).

Ki67 Index: On the 7th day, the Ki67 index was uninterpretable due to the lack of EP staining (Figure 4A). On the 10th and 17th days, the Ki67 index was significantly lower in the Manuka honey group compared to the control (*p* < 0.001) (Figure 4D,G). By the 60th day, there was no significant difference between the groups (*p* = 0.36) (Figure 4J).

Iba1 Expression: On the 7th and 10th days, Iba1-positive cell counts were significantly lower in the Manuka honey group compared to the control group (*p* < 0.001 and *p* = 0.04, respectively) (Figure 4B,E). No significant differences were observed on the 17th or 60th days (*p* > 0.05) (Figure 4H,K).

Collagen Density: On the 10th and 17th days, collagen density was significantly higher in the Manuka honey group than in the control group (*p* = 0.02) (Figure 4F,I). No significant differences were detected on the 7th or 60th days (*p* > 0.05) (Figure 4C,L).

After the 60th day post-injury, the wounds in all groups had healed. However, using MTS and PRS, we noticed fewer collagen fibers (Figure 2O,P) than in the Manuka-honey-treated group (Figure 2R,S). We can say that the number of collagen fibers in the Manuka-honey-treated group was more similar to the histological picture of healthy skin (the negative control) (Figure 2A,B).

Using the immunohistochemical technique, we did not notice differences between the control (Figure 3V,X,Z) and the Manuka-honey-treated groups (Figure 3W,Y,ZY).

### 2.3. Measurement of the Reepithelialization Area (REA)

The measurements of reepithelialization were based on images from a single representative pig, chosen due to its alignment with the average healing dynamics observed across the experimental groups. Figure 5 presents photographic documentation and a graphical analysis of the reepithelialization dynamics in the control and Manuka-honey-treated groups.

The reepithelialization process was meticulously observed from the moment of thermal injury induction until the achievement of complete wound closure, spanning up to 32 days (Figure 5A–P). Initial assessments across all experimental setups confirmed identical burn wound areas, thereby establishing a baseline reepithelialization rate of 0% for all groups. Significant advancements in the healing process were observed in the group treated with Manuka honey, particularly noted during the early stages of recovery. Statistical analyses revealed a marked difference in the rates of reepithelialization between the Manuka honey group and the control group on the 7th day post-injury (*p* < 0.05), indicating an enhanced healing response facilitated by the Manuka honey treatment (Figure 5C,D). This accelerated healing trajectory persisted, with the Manuka honey group demonstrating significantly greater wound closure compared to the control group on the 10th day post-injury (*p* < 0.01) (Figure 5E,F). The difference in healing efficacy became even more pronounced over time, highlighting the therapeutic potential of Manuka honey in promoting rapid wound recovery. Subsequent observations up to the 32nd day consistently showed that the Manuka honey treatment group experienced a faster and more efficient reepithelialization process when compared to the control group. By the conclusion of the study period, all wounds in the Manuka honey group had achieved complete epithelialization, underscoring the superior healing capabilities of Manuka honey.

### 2.4. Bacteriological Analysis

An antibiogram was conducted but is not presented since the wounds exhibited only micro-colonization by saprophytes, including *Staphylococcus epidermidis*, *Corynebacterium species*, *Propionibacterium acnes*, and non-beta-hemolytic *Streptococcus species* (e.g., *Streptococcus anginosus* and *Streptococcus mitis*). Clinically, there were no signs of infection.

### 2.5. Complications

During the study, complications included rectal prolapse in three experimental animals. This was effectively managed with repositioning and purse-string sutures. One animal experienced a recurrence, which required multiple sutures. Additionally, a leg fracture in another animal was conservatively treated. No animals were excluded from the study, and all were successfully returned to their natural habitat after 60 days.

## 3. Discussion

Honey’s antimicrobial properties have been recognized for centuries [7,8,9,10,12,15]. Manuka honey, which was specifically used in this study, has superior antibacterial activity compared to regular honey, due to its high levels of methylglyoxal, bee-defensin 1, and melanoids. In this study, Manuka honey was incorporated into a calcium alginate dressing. The alginate acts as a structural carrier, absorbing exudate, while Manuka honey delivers its bioactive compounds to promote wound healing by reducing the bacterial load and inflammation [8,10,19,20].

The pathohistological analysis on day 3 showed similar necrosis thickness values between the control and Manuka honey groups (662 μm vs. 657 μm). Early reepithelialization rates were also comparable (2.94% in the Manuka honey group vs. 2.42% in the control). However, by day 7, the Manuka honey group demonstrated a significantly higher reepithelialization rate (54%) compared to the control (31%). By day 10, reepithelialization reached 85% in the Manuka honey group, which was significantly higher than the control group (72%). Pancytokeratin (AE1/AE3) staining confirmed epithelial formation. Ki67 expression on day 7 was higher in the Manuka honey group compared to the control group, which could indicate an earlier onset of cell proliferation; however, this difference was not statistically significant. By days 10 and 17, Ki67 levels were significantly lower in the Manuka honey group (*p* < 0.001), suggesting a faster transition from the proliferative to the remodeling phase [21,22,23].

Reepithelialization is crucial in wound healing as it restores the skin’s barrier function, preventing infection and fluid loss. Manuka honey’s ability to accelerate this process likely results from its antibacterial and anti-inflammatory properties, which create an optimal environment for epithelial cells to proliferate and migrate. The therapeutic effects of Manuka honey on wound healing are attributed to its synergistic antimicrobial, anti-inflammatory, and cell-proliferative properties. The antimicrobial activity, primarily driven by methylglyoxal (MGO), reduces bacterial colonization and creates a sterile environment conducive to healing. Concurrently, its anti-inflammatory properties suppress pro-inflammatory cytokines, such as TNF-α and IL-6, reducing local tissue damage and promoting a faster transition to the proliferative phase [7,8,11]. This transition is further supported by Manuka honey’s ability to stimulate keratinocyte and fibroblast proliferation, which accelerates reepithelialization and extracellular matrix deposition. These mechanisms work in concert to create an optimal environment for wound healing, as evidenced by the accelerated reepithelialization observed in this study. The faster development of epithelial tissue, seen as early as day 7, points to Manuka honey’s role in promoting the growth factors necessary for epithelial proliferation. This faster reepithelialization, combined with reduced macrophage activity, suggests that Manuka honey improves healing by reducing prolonged inflammation and enhancing organized tissue regeneration [24,25,26]. Although the experimental conditions were standardized, external factors such as variations in animal stress, environmental conditions, or minor fluctuations in bacterial colonization could not be entirely eliminated. These variables were minimized through consistent housing, aseptic handling, and the close monitoring of the animals. The observed relationship between reduced inflammation and accelerated healing in the Manuka-honey-treated group further highlights the treatment’s robustness against such potential influences.

Japanese researchers [27] assessed the effect of Manuka honey on Jackson’s zone of stasis in an experiment on rats. The study did not prove that Manuka honey could affect the depth of the burn wound but showed significantly faster reepithelialization compared to the group of rats treated with silver sulfadiazine. This research aligns with our results.

Macrophages are essential in the inflammatory phase of wound healing, clearing pathogens and dead cells [28]. The quantitative analysis of Iba1 expression confirmed a significant reduction in macrophage activity in the Manuka honey group on days 10 and 17 (*p* < 0.05), supporting the hypothesis that this treatment promotes a faster resolution of inflammation. This reduction suggests that Manuka honey accelerates the resolution of inflammation, facilitating the transition to the proliferative phase [29]. By minimizing prolonged inflammation, Manuka honey contributes to more efficient tissue repair and reduces the risk of excessive scarring.

Histological staining, including Masson’s trichrome and Picrosirius red, revealed that the granulation tissue was thinner and the collagen density was higher in the Manuka honey group on days 10 and 17 (*p* < 0.05). However, further studies are required to quantitatively assess collagen fiber organization.

These findings suggest that Manuka honey stimulates fibroblast activity, enhancing collagen production and dermal regeneration. Fibroblasts are key players in wound healing, responsible for producing collagen and other extracellular matrix components that restore tissue structure [8,29,30,31]. The increased dermal density observed in the Manuka honey group points to improved dermal remodeling, a critical factor in reducing scar formation. This result is consistent with findings from studies like that of Ranzato et al., which demonstrated enhanced fibroblast function in vitro with Manuka honey [26].

Additionally, clinical observations revealed that Manuka honey promoted faster wound debridement, with necrotic tissue separating more quickly than in the control group [32]. Studies by Budak and Çakıroğlu [33] support this, showing that Manuka honey increases cell division, leading to faster reepithelialization, as evidenced by Ki-67 immunohistochemistry in mice models.

Interestingly, wounds with clinically better-formed epithelia showed a thinner epidermis (less than 80 µm), reduced epidermal ridges, and an improved reticular dermis structure. This observation suggests that a slightly thinner epidermis may indicate better maturation and a higher quality. Similar findings were confirmed in primary pilot studies, in which the structure of uninjured skin was compared in treated and control groups. These results highlight that the observed thinner epidermis should not be interpreted as incomplete healing but rather as a marker of improved tissue maturation.

A microbiological study was performed, and although no pathogenic bacteria were detected, only saprophytic organisms were found. This aligns with Manuka honey’s known antimicrobial properties [34,35,36,37,38], which likely contributed to the absence of infection signs. Previous studies have shown that Manuka honey is effective against both Gram-positive and Gram-negative bacteria, including resistant strains, and reduces biofilm formation, as demonstrated by Alandelajniet al. [39].

In burn treatment, managing superficial and deep dermal wounds is well established, but intermediate-depth burns, which combine features of both, pose a greater challenge [40]. Our study used FLIR One thermography on the first post-intervention day to categorize burns based on healing times. Superficial burns healed within 14 days, intermediate burns between 14 and 21 days, and deep dermal burns took over 21 days to heal. The temperature difference between the intact skin and deep dermal burns was 2.66 °C, indicating the need for surgical intervention. FLIR One thermography proved to be a non-invasive, cost-effective tool that could aid in treatment decisions, particularly for mixed-depth burns [41,42].

Although the primary focus of this study was not on wound temperature differences, the use of FLIR One thermography provided additional insights into burn depth variability. Variations in porcine skin thickness across different anatomical sites led to the classification of burns into both deep dermal and mesodermal categories.

Mesodermal burns exhibited a lower temperature difference compared to deep dermal burns (1.60 ± 0.34 °C vs. 2.66 ± 0.27 °C), reflecting differences in inflammatory activity and wound severity. This non-invasive method proved useful for identifying intermediate-depth burns, which are particularly challenging to manage. While exploratory in nature, these findings highlight the potential utility of FLIR thermography in categorizing burns and guiding treatment decisions in future studies.

Despite the promising findings, this study has certain limitations. The use of a porcine model, although closely mimicking human skin, may not fully replicate the complexities of human burn wound healing. Additionally, the sample size was limited, and further studies with larger cohorts are warranted to validate the findings. Future research should explore the long-term effects of Manuka honey on scar maturation and assess its efficacy for different types and severities of burns. Clinical trials involving human subjects are essential to confirm its translational potential and optimize application protocols.

The lack of additional controls such as a group with untreated burns or a vehicle-only group limits the study’s ability to isolate the specific effects of Manuka honey. Future research will incorporate these groups to address this limitation and provide a more comprehensive analysis of its therapeutic potential.

## 4. Materials and Methods

All procedures, including dressing changes and sample collection, were performed by personnel wearing protective clothing, such as gloves, masks, and sterile gowns, to maintain aseptic conditions. To prevent displacement of the dressing material and contamination of the wounds, a protective bandage suit was placed over the animals, ensuring the dressing remained secure and minimizing potential disruptions during the healing process.

### 4.1. Ethics Statement

The study was conducted entirely from an experimental perspective. All animal experiments were performed according to the European Directive for Protection of the Vertebrate Animals used for Experimental and Other Scientific Purposes 86/609/EES and the Principles of Good Laboratory Practice. Study approval was obtained from the Ministry of Agriculture and Environmental Protection (Belgrade, Serbia; No. 323-07-04449/2021-05).

In accordance with international ethical standards, we ensured the safety of the experimental model prior to the main study. A pilot test was conducted on two pigs to evaluate the potential toxicity of Manuka honey, and no local or systemic adverse effects were observed. Furthermore, the non-toxic nature of Manuka honey has been confirmed in previous studies [9,11], supporting its safe application for wound healing. The findings of the pilot test provided a basis for the main experimental procedures, ensuring compliance with OECD guidelines for animal studies.

### 4.2. Animals

Nine healthy female experimental animals (pigs) of the Landrace breed with a body mass of 25–30 kg, aged between two and three months, and weaned were randomly selected for the experiment. The initial examination was conducted at the Institute of Livestock Belgrade-Zemun in a clinic designated for examining and treating experimental animals. Before starting the experiment, a thorough assessment was performed to ensure that the animals did not have any health issues that would prevent the experiment from being carried out. This included checking for any systemic diseases or other comorbidities. Seven days before the experiment began, the animals were placed in individual cages with unrestricted access to food and water. The animals were kept in a room ranging from 20 to 25 °C, with air humidity of 55 ± 1.5% and a 12 h light–dark cycle. No animals developed any other comorbidity or burn disease after the burn injuries, so initially, no animals were automatically excluded from the study and they did not need to be treated by the veterinarian in charge of the welfare of the experimental animals.

### 4.3. Selection of Topical Agent

The preparation used in this study is an antibiotic ointment called Neosporin^®^ (*bacitracin zinc*, *neomycin sulfate*, and polymyxin B sulfate produced by Johnson & Johnson Consumer Inc., Skillman, NJ, USA). This combination is considered the gold standard for treating pediatric and adult populations, as bacitracin targets Gram-positive bacteria, neomycin targets both Gram-positive and Gram-negative bacteria, and polymyxin B targets Gram-negative bacteria. The use of antibiotic ointments did not disrupt epithelialization. Neosporin^®^ ointment (Johnson & Johnson Consumer Inc., Skillman, NJ, USA)contains the following active ingredients: bacitracin zinc: approximately 0.04%; neomycin sulfate: approximately 0.35%; and polymyxin B sulfate: approximately 0.5%. Algivon plus^®^ (Advancis Medical, Nottingham, UK) is an alginate dressing that was impregnated with a slow release of 100% Manuka honey whilst maintaining the integrity of the dressing. The Manuka honey used in this study was Algivon Plus (Advancis Medical), a product certified for medical use. It contains a standardized methylglyoxal (MGO) level of 400+, a pH of approximately 3.5, and bioactive compounds such as bee-defensin 1 and melanoids, which are known to enhance its antibacterial, anti-inflammatory, and wound healing properties.

Algivon Plus was selected for the following reasons:Medical Certification: It ensures the product is suitable for clinical use;Standardized Composition: Consistent levels of bioactive compounds ensure reproducible outcomes in wound care studies;Availability: Algivon Plus is widely accessible as a standardized product for research and clinical applications.

To provide a broader context, Table 2 summarizes other commercially available Manuka honey products and their characteristics.

### 4.4. Wound Model

The animals underwent general anesthesia with Diazepam Sopharma^®^—diazepam at a dose of 1.1 mg/kg (Sopharma AD, Sofia, Bulgaria) and Vetaketam^®^—ketamine at a dose of 15 mg/kg (VET-AGRO, Lublin, Poland) applied intramuscularly to the neck. Then, the skin on the back was shaved. After that, the operative field was prepared with an aerosol and was not rubbed to avoid causing skin hyperemia, which can affect the depth of the inflicted burn. Then, contact burns were applied with a brass attachment of a heater heated to 92 °C in contact with the skin for 15 s (Figure 6A,B). The burn depths in this study were standardized to simulate partial-thickness and deep dermal burns, consistent with clinical classifications of second- and third-degree burns. These classifications were informed by prior studies, including that of Wardhana et al., which reviewed methodologies for creating burn porcine models. Based on their recommendations, a brass template heated to 92 °C was applied for 15 s to induce deep dermal burns, aligning with validated methods for achieving reproducible and clinically relevant burn depths. The methodology was further supported by histological assessments to confirm burn depth consistency across experimental groups [43]. A total of 8 burns were formed on each pig (with dimensions of 47 mm by 47 mm), with four on the left side and four on the right side of the back. The burns were 20 mm apart and 30 mm from the spinal column to have approximately the same dermis thickness (Figure 6D). In total, there were 72 burned surfaces across nine animals. The burn wounds were divided into fields numbered 1, 3, 5, and 7. Each pig received treatment with Manuka honey, representing the *Manuka honey group*. Conversely, fields numbered 2, 4, 6, and 8 were treated with a combination of antibiotic ointments, representing the *control group*. After photographing the burns, each one was covered with a transparent polyurethane film, over which several layers of gauze were placed and positioned with wide circular bandages. The bandage placed in this way was protected with a protective coating. Additionally, each pig was housed in a cage of approximately 4 square meters to prevent injuries from the other animals. For the initial seven days following the application of the burn wounds, postoperative pain relief was administered using *Aanalgin*^®^-metamizole sodium (Alkaloid, Skopje, North Macedonia) at a dose of 25 mg/kg, administered via intramuscular injection. Deep dermal burns are characterized by a consistent whitish color in the central area, accompanied by a peripheral trail and surrounding redness, which were observed during a clinical examination.

On the 3rd, 7th, 10th, 14th, 17th, 20th, 23rd, and 30th day from the initiation of the burns, the animals were changed to the aforementioned topical treatment (Algivon plus and antibiotic ointments) with photo documentation of wounds. Before topical treatment, we performed smears of wounds for bacteriological analysis.

### 4.5. Infrared Camera (FLIR One Pro)

An *FLIR One Pro* infrared camera (FLIR, Täby, Sweden) connected to a smartphone was used for thermography during bandaging. The temperature difference between the tissue in the burned area and the undamaged skin was measured on the 1st day after intervention and noted (Figure 5E).

### 4.6. Tissue Samples’ Biopsy

Skin biopsies were taken using a 3 mm skin biopsy puncher on the 3rd, 7th, 10th, 17th, 20th, 23rd, and 60th days (Figure 5C). The biopsy on the 3rd day was used to determine the depth of the burn wound. Biopsies from the 7th to the 23rd day were taken to monitor the inflammatory and proliferative phases of healing. On the 60th day, biopsies were taken to evaluate the maturation phase.

### 4.7. Histological Tissue Processing

Tissue samples obtained with a bioptome were dehydrated through increasing alcohol concentrations (70, 80, 95, and 100%) and cleared with xylene, then embedded in paraffin. Paraffin molds were cut on a rotatory microtome (Sakura, Tokyo, Japan) to a thickness of 5 μm. All slides were stained using hematoxylin–eosin (H&E), Masson’s trichrome (MTS), and Picrosirius red (PRS) histochemical staining and the following immunohistochemical markers: rabbit anti-AE1/AE3 in a 1:50 dilution (Lab Vision; Thermo Scientific, Rockford, IL, USA); rabbit anti-Ki67 in a 1:300 dilution (Abcam; Cambridge, UK); and rabbit anti-Iba1 in a 1:8000 dilution (Abcam; Cambridge, UK). For visualization, we used mouse and rabbit peroxidase/DAB detection IHC kit from EnVision Detection Systems (DAKO Agilent; Santa Clara, CA, USA). A retrieval reaction included treatment with citrate buffer (pH 6.0) for 30 min at 99° C. Antibodies were applied for 60 min at room temperature with Mayer’s hematoxylin counterstain and finally mounted using DPX medium (Sigma-Aldrich, Steinheim, Germany). Slide analysis and digitalization were performed using a VisionTek^®^ digital microscope (Sakura, Japan).

### 4.8. Morphometric Analysis

#### 4.8.1. Measurement of the Reepithelialization Area (REA)

We capture high-resolution images of burns every time we dressed them using a Canon EOS 1200d camera (Canon, Tokyo, Japan). Using the free computer software Fiji 2.3.0 (Japan) and its plug-in (area), we measured the reepithelialized skin area (REA) (Figure 6F,G). REA was presented as the percentage of the reepithelialized part of the skin to the still unrecovered part, multiplied by 100%.

To calculate the Ki67 and Iba1 index, we took 20 representative microphotographs at 400× magnification per experimental group. Ten pictures were from the basal epidermal portion (EP), and ten were from the dermal portion (DP). For the Iba1 index, only the dermal portion (DP) was analyzed. We calculated the number of positive and negative cells using Fiji software. The value is presented as a percentage.

The collagen density was assessed using the Fiji (ImageJ 2.3.0) software. Microscopic images of Picrosirius red (PSR)-stained sections in TIFF format were processed through color deconvolution, selecting the “pink” mode to isolate the collagen signal. Calibration was performed by setting the upper threshold to 0 and the lower threshold to 100 (Figure 6H,I). Collagen density was calculated for each image and specimen, focusing on the entire organized granulation tissue. This methodology aligns with established protocols for quantifying collagen content in tissue sections. For instance, Kammerer et al. [44] developed a macro-based approach in Fiji for automated collagen quantification in PSR-stained heart sections, demonstrating the software’s applicability in such analyses.

#### 4.8.2. Measurement of Histological Morphometric Parameters

After digitizing histology slides using the free computer software Fiji (Tokyo, Japan) and its plug-in (distance), we measured the values of the necrotic debris thickness, epidermal thickness, epidermal ridge thickness, granulation tissue thickness, and reticular dermis thickness. For each digitalized slide, we performed three measurements, which are expressed as a mean value and presented in micrometers (µm).

### 4.9. Statistical Analysis

Descriptive statistical methods and hypothesis testing were used to analyze the primary data. Descriptive methods included measures of central tendency (arithmetic mean) and variability (standard deviation), along with relative frequencies (structure indicators).

For hypothesis testing, both parametric and non-parametric statistical analyses were applied, depending on the nature of the data:Normality of data distribution was assessed using the Shapiro–Wilk test;Parametric tests, such as the Student’s t-test, were used for group comparisons in which data followed a normal distribution;Non-parametric tests, including the Mann–Whitney U test, were employed for non-normally distributed data.

Specific statistical methods were applied to the following datasets:Reepithelialization rates: The Mann–Whitney U test was used to compare the experimental (Manuka honey) and control (antibiotic ointment) groups at different time points;Granulation tissue thickness: Differences between groups over time were assessed using the Mann–Whitney U test;Collagen density and orientation: Student’s t-test was conducted for between-group comparisons, in which normality was confirmed.

All hypotheses were tested at a significance level of α = 0.05. Results are presented in tables, graphs, and microphotographs to provide a comprehensive understanding of the findings. Data analysis was performed using IBM SPSS Statistics 26, and charts and tables were created with Microsoft Office Word 2007.

## 5. Conclusions

In conclusion, this study suggests a potential benefit of applying Manuka honey in alginate for burn wound healing compared to standard antibiotic ointments. The observed effects may be attributed to the antimicrobial and antioxidant properties of Manuka honey, as well as its hydrogel formation, which potentially creates a favorable environment for wound repair. Although burns treated with Manuka honey appeared to show some cosmetic and pathohistological improvements, such as reduced granulation tissue and a thinner epidermis, these findings are preliminary and require further validation through more comprehensive studies.

The quantitative analysis revealed that the Manuka-honey-treated wounds exhibited significantly lower macrophage activity (Iba1) and reduced Ki67 expression on days 10 and 17, suggesting a faster transition to the tissue remodeling phase. Additionally, the collagen density was significantly higher in the Manuka honey group on days 10 and 17, indicating enhanced extracellular matrix deposition. However, further studies are required to assess collagen fiber organization and confirm these structural differences.

The faster reepithelialization observed in the Manuka honey group indicates a potential positive impact on the healing environment, but this observation must be interpreted cautiously due to the absence of additional control groups in this study. Future research should address these limitations and include control groups with healthy skin, untreated burns, and the vehicle only to isolate the specific effects of Manuka honey.

The study also mentions the potential use of forward-looking infrared (FLIR) thermography as a tool for assessing burn wound depth. Still, additional research is needed to confirm its clinical utility. We acknowledge that Algivon is a well-known product, and no conflict of interest exists related to its use in this study.

## Figures and Tables

**Figure 1 pharmaceuticals-18-00296-f001:**
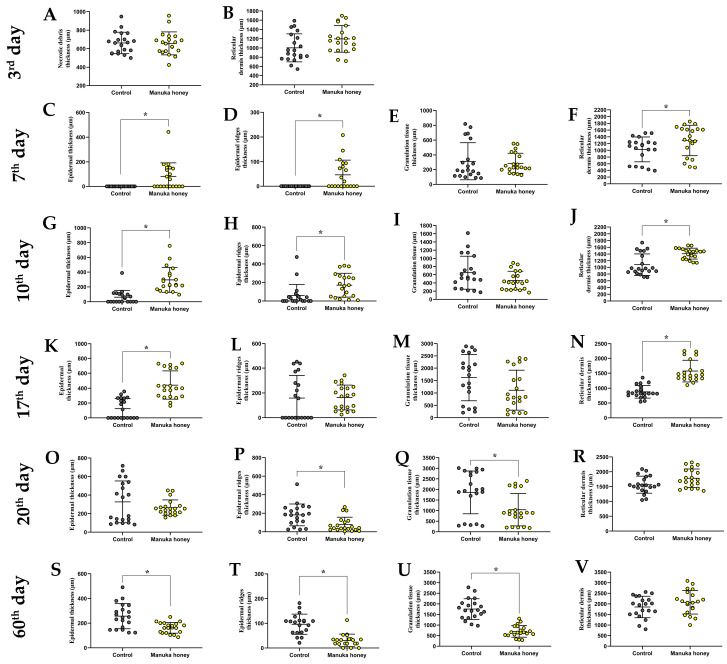
Morphometric analysis of skin compartments: necrotic debris thickness (just on third day); epidermal thickness; epidermal ridge thickness; granulation tissue thickness; reticular dermis thickness. Subfigures (**A**–**V**) represent measurements at different time points: 3rd, 7th,10th,17th,20th, 60th day. The asterisk (*) denotes statistically significant differences (*p* < 0.05).

**Figure 2 pharmaceuticals-18-00296-f002:**
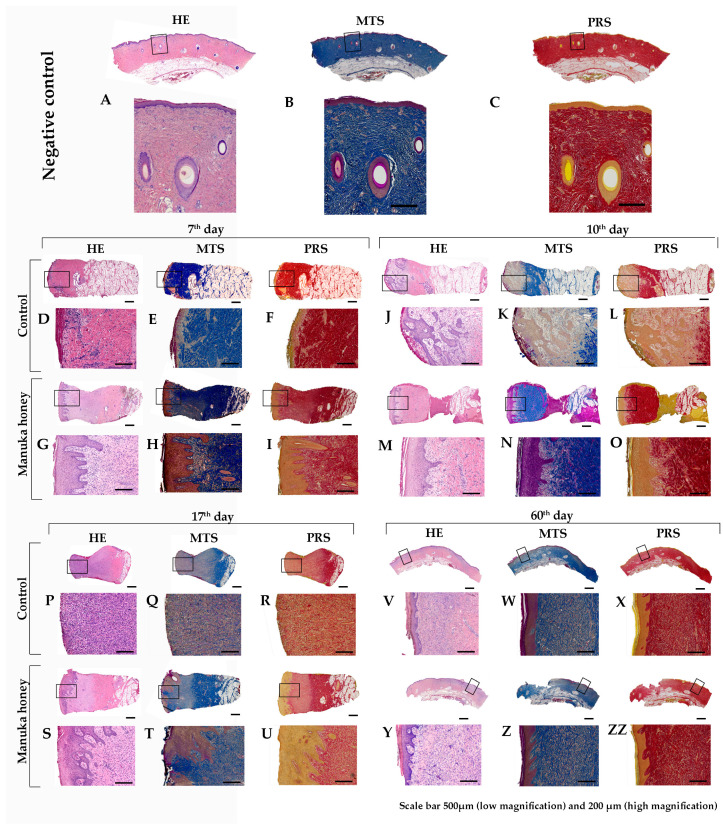
Histochemical staining of wound lesions. Scan of whole-tissue biopsy with magnification of the black framed area. (**A**,**B**) Representative healthy normal porcine skin (negative control group). (**C**–**ZZ**) show histochemical staining of wound biopsies at different post-injury time points using Masson’s trichrome (MTS) and Picrosirius red (PSR) staining techniques: (**C**–**I**) 7th day, (**J**–**O**) 10th day, (**P**–**U**) 17th day, and (**V**–**ZZ**) 60th day. Each time point includes images from both the control and Manuka honey-treated groups, highlighting differences in collagen deposition and tissue remodeling. The larger images represent whole-tissue scans, while the black-framed areas indicate regions selected for high-magnification microphotographs. Magnification of scan (scale bar 500 µm) and microphotograph (scale bar 200 µm).

**Figure 3 pharmaceuticals-18-00296-f003:**
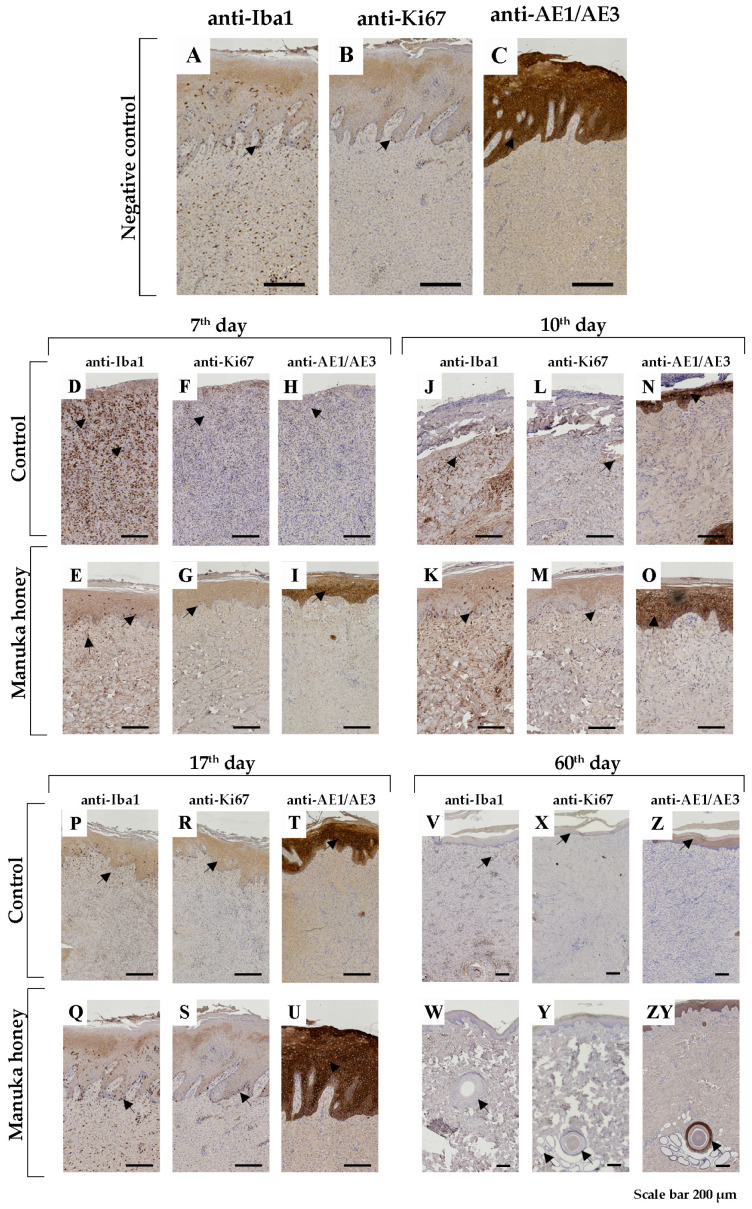
Immunohistochemical (IHC) staining of wound lesions. Macrophage marker—Iba1 (asterisk—cytoplasmic positivity); Proliferative marker—Ki67 (arrowhead—nuclear positivity); Pan-cytokeratin—AE1/AE3 (arrowhead—cytoplasmic positivity); (**A**–**C**) Representative IHH staining (anti-Iba1, anti-Ki67, and anti-AE1/AE3, respectively); healthy normal porcine skin (negative control group). (**D**–**ZY**) show IHC staining of wound lesions at different time points post-injury in the control and Manuka-honey-treated groups. (**D**–**I**) 7th day, (**J**–**O**) 10th day, (**P**–**U**) 17th day, and (**V**–**ZY**) 60th day. Scale bar: 200 µm.

**Figure 4 pharmaceuticals-18-00296-f004:**
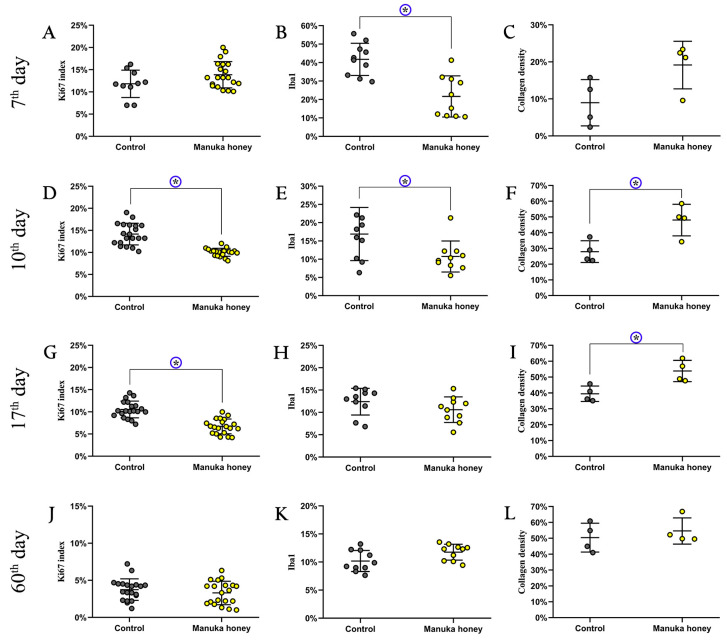
Effect of manuka honey on wound healing: proliferation, inflammation, and collagen deposition over time. Quantitative analysis of wound healing parameters at different time points (7th, 10th, 17th, and 60th day) in the control and Manuka-honey-treated groups (**A**,**D**,**G**,**J**). Ki67 proliferation index, indicating cellular proliferation (**B**,**E**,**H**,**K**). Iba1 expression, representing macrophage infiltration (**C**,**F**,**I**,**L**). Collagen density, reflecting extracellular matrix deposition. Data are presented as mean ± standard deviation. Statistically significant differences between groups are marked with an asterisk (*).

**Figure 5 pharmaceuticals-18-00296-f005:**
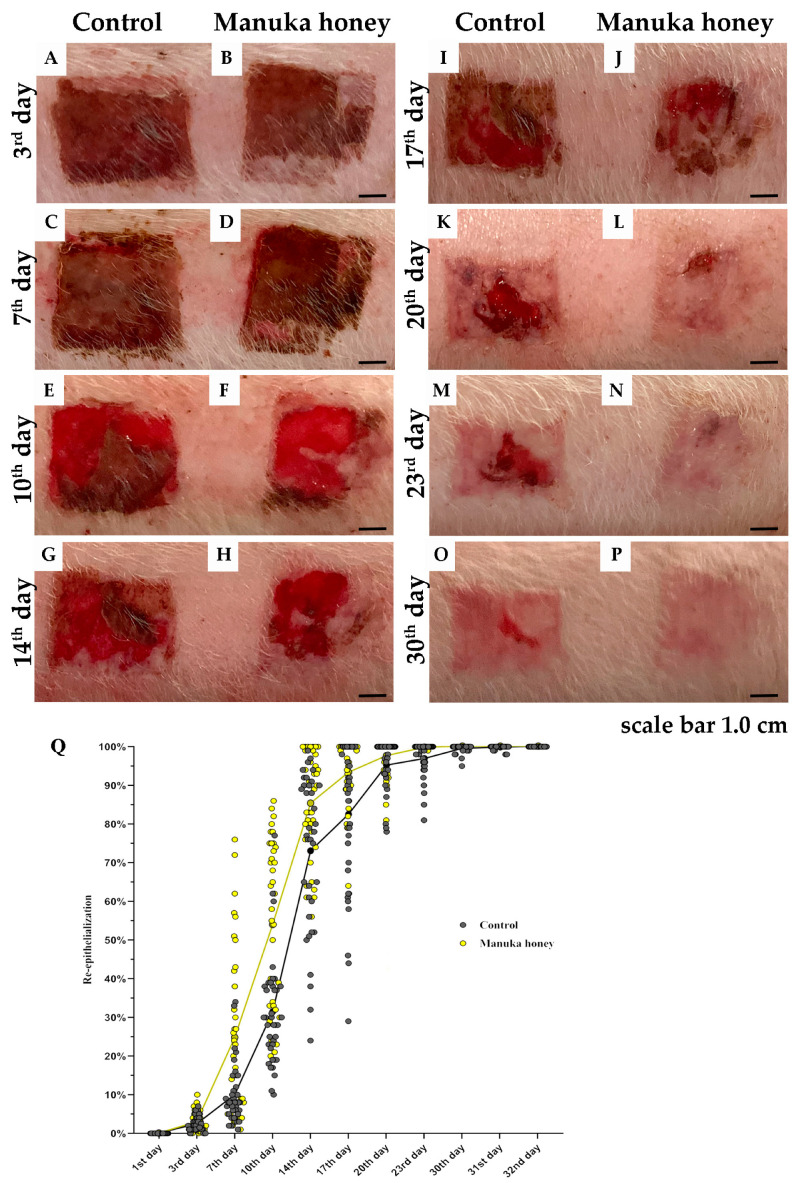
Measurement o f the reepithelialization area (REA). Photo documentation of wound healing progression in control and Manuka honey-treated groups based on images from a single representative pig. (**A**–**P**) show macroscopic wound images at different time points: (**A**,**B**) 3rd day, (**C**,**D**) 7th day, (**E**,**F**) 10th day, (**G,H**) 14th day, (**I**,**J**) 17th day, (**K**,**L**) 20th day, (**M**,**N**) 23rd day, and (**O**,**P**) 30th day. (**Q**) Graphical representation of REA dynamics for both groups, illustrating the differences in wound healing progression. Scale bar: 1.0 cm.

**Figure 6 pharmaceuticals-18-00296-f006:**
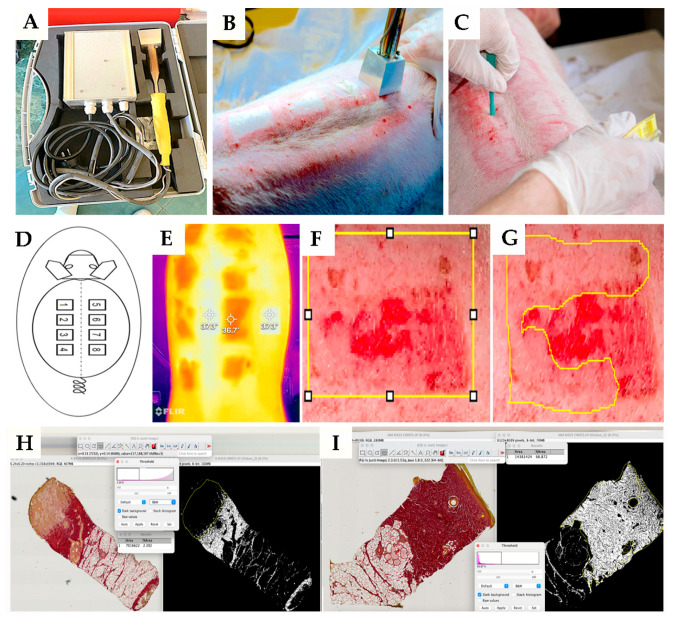
Illustrations of the experimental procedure. Heater and its application (**A**,**B**); technique of tissue sampling (**C**); schema of burn place application (**D**); representative photograph of skin and wound using an infrared camera (**E**); measurement of the whole area (yellow borders on picture (**F**)) and reepithelialization area (yellow borders on picture—(**G**)) of the wound on 30th day post-thermal injury; measurement and quantification of collagen density in wound tissue using Picrosirius red staining (**H**,**I**).

**Table 1 pharmaceuticals-18-00296-t001:** Mean and standard deviation values of the temperature difference in the wound area (°C).

Reepithelialization (Time)	Mean Value (°C)	Standard Deviation (SD)
Mesodermal burns	1.60	0.34
Deep dermal burns	2.66	0.27

**Table 2 pharmaceuticals-18-00296-t002:** Overview of commercially available Manuka honey products and their key characteristics.

Product Name	MGO Level (mg/kg)	Certification Type	Manufacturer	Notes
Algivon Plus Manuka Honey	400+	Medical certification	Advancis Medical	Designed for clinical use.
Medihoney Antibacterial Honey	350+	Medical certification	Derma Sciences	Used for wound treatment and debridement.
Comvita Manuka Honey UMF 10+	263+	UMF certification (UMFHA)	Comvita	Commercially available for consumption.
Manuka Health MGO 550+	550+	MGO certification	Manuka Health	Suitable for dietary and medical use.

## Data Availability

The raw data supporting this article’s conclusions are available upon request.

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
