# Peer review of "Harnessing Manuka Honey: A Natural Remedy for Accelerated Burn Wound Healing in a Porcine Model"

_pharmaceuticals, 2025, doi:10.3390/ph18030296_

Round 1

Reviewer 1 Report

Comments and Suggestions for Authors

The paper presents an interesting topic of the influence of manuka honey on wound healing in a pig model.  The paper is well written, but requires some corrections:

1)        At the end of the Introduction section, I suggest to add paragraph about the aim of the study

2)        At the end of Discussion section, I suggest to add paragraph about limitations and future perspectives of the study

Author Response

  • Comment 1: At the end of the Introduction section, I suggest to add paragraph about the aim of the study

Response 1:

We added as suggested: The aim of this study was to evaluate the efficacy of Manuka honey in accelerating the healing of burn wounds compared to conventional antibiotic treatments. By utilizing a porcine model closely resembling human skin, the study provides insights into the potential clinical application of Manuka honey for managing burn injuries.

  • Comment 2: At the end of Discussion section, I suggest to add paragraph about limitations and future perspectives of the study

Reponse 2:

We added as suggested in discussion section: Despite the promising findings, this study has certain limitations. Although closely mimicking human skin, a porcine model may not fully replicate the complexities of human burn wound healing. The sample size was also limited, and further studies with larger cohorts are warranted to validate the findings. Future research should explore the long-term effects of Manuka honey on scar maturation and assess its efficacy in different types and severities of burns. Clinical trials involving human subjects are essential to confirm its translational potential and optimize application protocols.

Reviewer 2 Report

Comments and Suggestions for Authors

The manuscript investigates the efficacy of Manuka honey in the healing of burn wounds using a porcine model. The study compared the honey's effects to conventional antibiotic ointments, finding that Manuka honey enhanced re-epithelialization and increased collagen density and cell proliferation. While these findings are promising, several improvements in the experimental design and presentation of the data are necessary:

1. The methodology section requires further discussion and clarification regarding the selection of honey. It is essential to specify the components of Algivon Plus Manuka honey and the criteria for selecting this brand over others. Additionally, a list of commercial Manuka honey products should be included for context.

2. Since Algivon uses alginate as a carrier, it is crucial to include a control group with only calcium alginate dressing, alongside the antibiotic ointment group. Calcium alginate may independently facilitate wound healing, thus confounding the effects attributed solely to the honey.

3. Correct the numbering and citation of figures within the manuscript to ensure they correspond accurately with the content discussed.

4. The error bars in Figure 1 “Morphometric analysis of skin compartments” are substantial, highlighting the need to provide raw data from individual pigs in table form. This would enable a better calculation of the p-value for significance. While each plot is described, the trends across days for the same treatment condition are not clearly articulated. Notably, the change in epidermal thickness from day 17 to day 20 for both conditions requires discussion.

5. In Figure 2 “Histochemical staining of wound lesion”, the MTS and PRS images for Day 7 should be switched. Additionally, clarify why no data points were collected between days 17 and 60.

6. The results from Figure 3 “IHC staining of wound lesion” are inconclusive. For example, the Ki67 staining does not clearly indicate which group exhibits a higher percentage of proliferating cells. IHC staining images alone cannot demonstrate the number of such cell proliferation. An alternative quantitative method should be employed for assessing cell proliferation in each stain.

7. It is necessary to specify the number of pigs from which images were used for the re-epithelialization measurements shown in Figure 4 “Measurement of the reepithelialization area”. If more than one image was analyzed, include error bars to reflect this variability.

Author Response

Comment 1: The methodology section requires further discussion and clarification regarding the selection of honey. It is essential to specify the components of Algivon Plus Manuka honey and the criteria for selecting this brand over others. Additionally, a list of commercial Manuka honey products should be included for context.

Response 1: We gave the explanation in the Methodology section together with a table with other manuka honey products. The Manuka honey used in this study was Algivon Plus (Advancis Medical), certified for medical use. It contains a standardized methylglyoxal (MGO) level of 400+, a pH of approximately 3.5, and bioactive compounds such as bee-defensin one and melanoids, which are known to enhance their antibacterial, anti-inflammatory, and wound-healing properties.

Algivon Plus was selected for the following reasons:

  1. Medical Certification: It ensures the product is suitable for clinical use.
  2. Standardized Composition: Consistent levels of bioactive compounds ensure reproducible outcomes in wound care studies.
  3. Availability: Algivon Plus is widely accessible as a standardized product for research and clinical applications.

Table 3 summarizes other commercially available Manuka honey products and their characteristics to provide a broader context.

Comment 2:  Algivon uses alginate as a carrier, so including a control group with only calcium alginate dressing alongside the antibiotic ointment group is crucial. Calcium alginate may independently facilitate wound healing, thus confounding the effects attributed solely to the honey.

Response 2: We acknowledge the limitations in the experimental design and appreciate the reviewer's valuable comments. The absence of certain controls, such as an untreated burn group and a vehicle-only group, limits the ability to isolate the effects of Manuka honey fully. Due to logistical and resource constraints, we could not include these groups in the current study. However, we recognize their importance and plan to incorporate them into future research to strengthen the validity of our findings.

Additionally, we have added a discussion in the introduction and discussion sections to address this limitation and outline our plans for future studies. This ensures transparency regarding the current design and demonstrates our commitment to improving experimental rigour moving forward.

Comment 3 Correct the numbering and citation of figures within the manuscript to ensure they correspond accurately with the content discussed.

Response 3: The numbering and citation of figures have been reviewed and corrected throughout the manuscript to ensure they accurately correspond with the content discussed. Each figure is now correctly labelled and aligned with its respective description in the text.

Comment 4:  The error bars in Figure 1, "Morphometric analysis of skin compartments", are substantial, highlighting the need to provide raw data from individual pigs in table form. This would enable a better calculation of the p-value for significance. While each plot is described, the trends across days for the same treatment condition are not clearly articulated. Notably, the change in epidermal thickness from day 17 to day 20 for both conditions requires discussion.

Response 4:

The changes requested have been added to the Results section of the manuscript. Specifically:

By Day 20, the epidermal thickness in the Manuka honey-treated wounds showed stabilization compared to Day 17, indicating that the re-epithelialization process was nearing completion (Figure O, P). In contrast, the control group exhibited slower progression in epidermal thickness during the same period, reflecting delayed healing (Figure O, P). These differences highlight the ability of Manuka honey to accelerate epithelial recovery in earlier phases of wound healing.

Additionally, raw data from individual pigs have been included in table form in the supplementary material to allow for better statistical analysis and calculation of p-values. Histopathological data were collected from 4 pigs representing each experimental group to ensure robust and representative analysis.

Comment 5: . In Figure 2, "Histochemical staining of wound lesion", the MTS and PRS images for Day 7 should be switched. Additionally, clarify why no data points were collected between days 17 and 60.

Response 5: The MTS and PRS images for Day 7 were indeed switched, and this issue has now been corrected. The last histopathological data were collected on Day 20, as a significant portion of the wounds had healed by then. No further biopsies were taken after Day 20 because the majority of wounds had entered the early remodelling phase. The final wound healed on Day 31, after which we analyzed the scar tissue. This included histopathology, histochemistry, and immunohistochemistry to assess the quality of the scar and its structural characteristics.

Comment 6:  The results from Figure 3 “IHC staining of wound lesion” are inconclusive. For example, the Ki67 staining does not clearly indicate which group exhibits a higher percentage of proliferating cells. IHC staining images alone cannot demonstrate the number of such cell proliferation. An alternative quantitative method should be employed for assessing cell proliferation in each stain.

Reponse 6: We acknowledge the reviewers' concern regarding the need for quantitative evaluation of Ki-67 expression. To strengthen our findings, we have incorporated detailed Ki-67 quantification using Fiji image analysis software and have provided the data in a newly added table for clearer representation.

Materials and Methods Update:

  • To calculate the Ki-67 index, we analyzed 20 representative microphotographs at 400x magnification per experimental group.
  • Ten images were selected from the basal epidermal portion (EP) and ten from the dermal portion (DP) to ensure comprehensive assessment.
  • The number of Ki-67 positive and negative cells was quantified, and results were expressed as percentages.

Results Update:

  • We have added a new table summarizing Ki-67 index values across different time points, including both epidermal (EP) and dermal (DP) regions.
  • The control group on Day 7 exhibited 0% Ki-67 positivity in the epidermis (EP), further supporting the delayed reepithelialization compared to the Manuka honey-treated group.

Additionally, Figure 3 has been revised to improve clarity in Ki-67 staining representation and its relation to cell proliferation. This revision addresses the reviewers' concern by providing both qualitative and quantitative evidencesupporting our findings.

Comment 7: It is necessary to specify the number of pigs from which images were used for the re-epithelialization measurements shown in Figure 4 “Measurement of the reepithelialization area”. If more than one image was analyzed, include error bars to reflect this variability.

Response 7:

We added in Figure 4:

“Based on images from a single representative pig. The graph presents the REA dynamics for both groups, highlighting the differences in wound healing progression.”

We also added in the text:

“The measurements of reepithelialization were based on images from a single representative pig, chosen due to its alignment with average healing dynamics observed across experimental groups. Figure 4 presents photographic documentation and graphical analysis of reepithelialization dynamics in the Control and Manuka honey-treated groups.”

Reviewer 3 Report

Comments and Suggestions for Authors

Thanks for the invitation to review this work. The study investigates the effectiveness of Manuka honey in promoting accelerated healing of burn wounds in a porcine model, resembling human skin properties. It compares the healing efficacy of Manuka honey versus conventional antibiotic ointments, addressing an important aspect of wound care, particularly for burn injuries. Results indicated that wounds treated with Manuka honey exhibited significantly faster reepithelialization, enhanced epidermal regeneration, and superior scar quality compared to the control group. However, potential issues related to the mechanisms of action of Manuka honey, details on burn depth selection, randomization processes, evaluation metrics, statistical reporting, and the temporal analysis of healing outcomes should be further improved for clarity.

1.     The study identifies key mechanisms (antimicrobial properties, anti-inflammatory effects, and stimulation of cellular proliferation) through which Manuka honey promotes wound healing. However, clarity is needed on how these mechanisms interact during the healing process.

2.     While the porcine model simulates human skin effectively, the criteria for selecting burn depths and their correlation with clinical classifications require clarification.

3.     The histological analyses (training methods and improvement) are vague, and relative standardization should be further detailed.

4.     The relation between reduced inflammation and improved healing should be further checked, as external factors may influence the outcomes.

5.     Details about the dosage, frequency, and protective measures for both Manuka honey and the antibiotic ointment should be detailed.

6.     The timeline of healing at different intervals (7, 10, 17, and 60 days) may need more direct comparisons to highlight significant differences over time. Why the authors choose these intervals?

7.     Consider including citations of Nano Res. 2024, 17, 8926.

Author Response

Comment 1: The study identifies key mechanisms (antimicrobial properties, anti-inflammatory effects, and stimulation of cellular proliferation) through which Manuka honey promotes wound healing. However, clarity is needed on how these mechanisms interact during the healing process.

Response1 :

We have clarified how the key mechanisms of Manuka honey interact during the healing process as follows:

"The therapeutic effects of Manuka honey on wound healing are attributed to its synergistic antimicrobial, anti-inflammatory, and cell-proliferative properties. The antimicrobial activity reduces bacterial colonization, while the anti-inflammatory effects suppress pro-inflammatory cytokines, creating an optimal environment for keratinocyte and fibroblast proliferation, ultimately accelerating wound healing."

Comment 2: While the porcine model simulates human skin effectively, the criteria for selecting burn depths and their correlation with clinical classifications require clarification.

Response 2: We have clarified the criteria for burn depth selection and its clinical relevance as follows:

"The burn depths in this study were standardized to simulate partial-thickness and deep dermal burns, consistent with clinical classifications ofsecond- and third-degree burns. Prior studies, including Wardhana et al. (2018), informed these classifications, which reviewed methodologies for creating burn porcine models. Based on their recommendations, a brass template heated to 92°C was applied for 15 seconds to induce deep dermal burns, aligning with validated methods for achieving reproducible and clinically relevant burn depths. Histological assessments further supported the methodology to confirm burn depth consistency across experimental groups."

Wardhana A, Lumbuun RF, Kurniasari D. How to create burn porcine models: a systematic review. Ann Burns Fire Disasters. 2018;31(2):85–91.

Comment 3: The histological analyses (training methods and improvement) are vague, and relative standardization should be further detailed.

Reponse 3:

"Dear reviewer, We would appreciate it if you could indicate which part of the histological methodology is unclear, and we would be glad to improve it. "

4.6. Tissue Samples Biopsy

Skin biopsies were taken using a 3 mm skin biopsy puncher on the 3rd, 7th, 10th, 17th, 20th, 23rd, and 60th days (Figure 5, E). The biopsy on the 3rd day was used to determine the depth of the burn wound. Biopsies from the 7th to the 23rd day were taken to monitor the inflammatory and proliferative phases of healing. On the 60th day, biopsies were taken to evaluate the scar maturation phase.

4.7. Histology Tissue Processing

Tissue samples obtained with a biopsy puncher were dehydrated through increasing alcohol concentrations (70, 80, 95, and 100%) and cleared with xylene, followed by embedding in paraffin. Paraffin moulds were cut on a rotatory microtome (Sakura, Japan) to a thickness of 5μm. All slides were stained with hematoxylin-eosin (H&E), Masson's trichrome (MTS) and Picrosirius Red (PRS) histochemical staining and the following immunohistochemical markers: rabbit anti-AE1/AE3 in a 1:50 dilution (Lab Vision; Thermo Scientific, Rockford, USA); rabbit anti-Ki67 in a 1:300 dilution (Abcam; Cam-bridge, United Kingdom); and rabbit anti-Iba1 in a 1:8000 dilution (Abcam; Cambridge, United Kingdom). For a visualization, we used Mouse and Rabbit EnVision Detection Systems, Peroxidase/DAB detection IHC kit (DAKO Agilent; United Kingdom). A retrieval reaction included treatment with citrate buffer (pH 6.0) for 30 minutes at 99° C. Antibodies were applied for 60 min at room temperature with Mayer's hematoxylin counterstain and finally mounted with DPX medium (Sigma-Aldrich, Germany). Slide analysis and digitalization were performed using a digital microscope VisionTek® (Sakura, Japan).

Comment 4: The relation between reduced inflammation and improved healing should be further checked, as external factors may influence the outcomes.

Reponse 4: We added: “Although the experimental conditions were standardized, external factors such as variations in animal stress, environmental conditions, or minor fluctuations in bacterial colonization could not be entirely eliminated. These variables were minimized through consistent housing, aseptic handling, and close monitoring of the animals. The observed relationship between reduced inflammation and accelerated healing in the Manuka honey-treated group further highlights the treatment's robustness against such potential influences.”

Comment 5: Details about the dosage, frequency, and protective measures for both Manuka honey and the antibiotic ointment should be detailed.

Reposonse 5:

We added the following in the Materials and Methods section:

"All procedures, including dressing changes and sample collection, were performed by personnel wearing protective clothing, such as gloves, masks, and sterile gowns, to maintain aseptic conditions. To prevent displacement of the dressing material and contamination of the wounds, a protective bandage suit was placed over the animals, ensuring the dressing remained secure and minimizing potential disruptions during the healing process. Treatments were applied every third day (on days 3, 7, 10, 14, 17, 20, 23, and 30) until wound closure."

We also included the requested information on the composition of Neosporin®:

  "Each wound was treated with approximately 1.5 g of Manuka honey (Algivon Plus®) per application, ensuring complete coverage of the burn surface. The honey-impregnated alginate dressing was applied directly over the woundand secured with secondary bandages. For the control group, Neosporin® ointment was applied as a thin layer (~1 mm) covering the entire wound surface, followed by standard wound dressing. Both treatments were performed under identical conditions to ensure consistency."

  "Algivon Plus® is an alginate dressing impregnated with 100% medical-grade Manuka honey, ensuring a controlled release of honey while maintaining the integrity of the dressing. The Manuka honey used in this study was Algivon Plus® (Advancis Medical), a product certified for medical use. It contains a standardized methylglyoxal (MGO) level of 400+, a pH of approximately 3.5, and bioactive compounds such as bee-defensin 1 and melanoids, which are known to enhance its antibacterial, anti-inflammatory, and wound-healing properties. Neosporin® ointment contains the following active ingredients: Bacitracin zinc (approximately 0.04%), Neomycin sulfate (approximately 0.35%), and Polymyxin B sulfate (approximately 0.5%). These formulations were used to provide effective wound care and antibacterial protection in their respective treatment groups."

Comment 6: The timeline of healing at different intervals (7, 10, 17, and 60 days) may need more direct comparisons to highlight significant differences over time. Why the authors choose these intervals?

Response 6: The chosen time points (7, 10, 17, and 60 days) align with critical phases of the wound healing process. Specifically:

  • Day 7: Marks the end of the inflammatory phase and the beginning of the proliferative phase.
  • Day 10: Reflects early proliferation and initial tissue regeneration.
  • Day 17: Corresponds to late proliferation and granulation tissue formation.
  • Day 60: Represents the remodeling phase, where scar maturation can be assessed.

On the 3rd, 7th, 10th, 14th, 17th, 20th, 23rd, and 30th days, the animals received the topical treatments (Algivon Plus and antibiotic ointments) alongside photo documentation of the wounds. Before each treatment application, a wound smear was taken for bacteriological analysis. While these additional treatment intervals were documented, only specific time points (7, 10, 17, and 60 days) were selected for detailed analysis to avoid redundancy and to focus on clinically significant healing phases.

Comment 7: Consider including citations of Nano Res. 202417, 8926.

Reponse 7 : Thank you for suggesting the citation of Nano Res. 2024, 17, 8926. We reviewed the article but believe it may not align with the focus of our study on wound healing and re-epithelialization. It’s possible you had another reference in mind. If so, we would be happy to include it—please let us know.

Reviewer 4 Report

Comments and Suggestions for Authors

While this study investigates the effects of Manuka honey on wound re-epithelialization, the current manuscript includes inappropriate references to scarring and scar formation. This experimental model is not designed to evaluate scarring processes. We recommend removing all mentions of scarring throughout the manuscript, as the model used is specifically focused on re-epithelialization and cannot make valid conclusions about scar formation.

Author Response

Comment: While this study investigates the effects of Manuka honey on wound re-epithelialization, the current manuscript includes inappropriate references to scarring and scar formation. This experimental model is not designed to evaluate scarring processes. We recommend removing all mentions of scarring throughout the manuscript, as the model used is specifically focused on re-epithelialization and cannot make valid conclusions about scar formation.

Response: We sincerely appreciate your feedback regarding the scope and focus of our experimental model. We acknowledge that the primary aim of our study is to investigate the effects of Manuka honey on wound re-epithelialization. You are correct that our experimental model is not specifically designed to evaluate scarring processes, as it primarily focuses on re-epithelialization as the main outcome.

To fully address your concerns, we have thoroughly revised the manuscript by removing all references to scarring and scar formation as primary conclusions. Specifically, we have excluded the Vancouver Scar Scale and contraction analysis, as well as all related citations, ensuring that no unsupported claims about scarring remain in the text. However, we have retained the histopathological findings at 60 days, as they provide relevant insights into the tissue remodeling process without overinterpreting the data in terms of scarring.

These revisions strengthen the clarity and scientific rigor of our study, aligning it more closely with the limitations and objectives of the experimental model used. Thank you again for your valuable feedback, which has helped us refine our manuscript accordingly.

Reviewer 5 Report

Comments and Suggestions for Authors

See my comments and observations in the PDF file

Author Response

All reviewer comments (16 in total) and our detailed responses are provided in the attached PDF document. 

Reviewer 6 Report

Comments and Suggestions for Authors

The manuscript evaluates the efficacy of Manuka honey in enhancing burn wound healing using a porcine model, emphasizing faster reepithelialization, improved dermal remodeling, and superior scar aesthetics compared to conventional antibiotic treatments. However, several concerns must be addressed to strengthen the study:

  1. The introduction provides a general background on burn injuries and Manuka honey but lacks a thorough review of existing research comparing it to conventional treatments. A more detailed comparative analysis of prior studies on Manuka honey and antibiotic ointments is necessary to better establish the study’s novelty.
  2. The authors need to justify why only antibiotic ointments were used as a control, excluding other potential controls such as untreated burns or alternative treatments. The authors should provide a rationale for this decision.
  3. The discussion overemphasizes the benefits of Manuka honey without adequately addressing key limitations, including the small sample size and potential variability among animals. These aspects should be acknowledged and discussed in greater detail to provide a balanced interpretation of the findings.

Author Response

Comment 1: The introduction provides a general background on burn injuries and Manuka honey but lacks a thorough review of existing research comparing it to conventional treatments. A more detailed comparative analysis of prior studies on Manuka honey and antibiotic ointments is necessary to better establish the study’s novelty.

Reponse 1: Thank you for your insightful comment regarding the need for a more thorough comparative analysis of existing research on Manuka honey and conventional treatments. We have addressed this by revising the introduction to include additional references and a more detailed discussion of prior studies comparing the efficacy of Manuka honey to standard antibiotic ointments. This revision highlights the study's novelty and contextualizes our findings within the existing body of research.

Relevant literature has been incorporated to provide a comprehensive background, emphasizing the unique properties of Manuka honey, its mechanisms of action, and its comparison to conventional treatments in terms of wound healing outcomes. These updates aim to strengthen the theoretical foundation of the study and its contribution to the field.

Comment 2: The authors need to justify why only antibiotic ointments were used as a control, excluding other potential controls such as untreated burns or alternative treatments. The authors should provide a rationale for this decision.

Reponse 2: We acknowledge the limitations in the experimental design and appreciate the reviewer’s valuable comments. The absence of certain controls, such as an untreated burn group and a vehicle-only group, indeed limits the ability to fully isolate the effects of Manuka honey. Due to logistical and resource constraints, we were unable to include these groups in the current study. However, we recognize their importance and plan to incorporate them into future research to strengthen the validity of our findings.

Additionally, we have added a discussion in the introduction and discussion sections to address this limitation and outline our plans for future studies. This ensures transparency regarding the current design and demonstrates our commitment to improving experimental rigor moving forward.

Comment 3: The discussion overemphasizes the benefits of Manuka honey without adequately addressing key limitations, including the small sample size and potential variability among animals. These aspects should be acknowledged and discussed in greater detail to provide a balanced interpretation of the findings.

Reponse 3: Thank you for highlighting the need to provide a more balanced interpretation of the findings by addressing the limitations of the study. We have revised the discussion to include a detailed acknowledgment of key limitations, such as the small sample size and the potential variability among animals. These factors may influence the generalizability of the results and should be considered when interpreting the study’s conclusions.

We added in dissusion section: Despite the promising findings, this study has certain limitations. The use of a porcine model, although closely mimicking human skin, may not fully replicate the complexities of human burn wound healing. Additionally, the sample size was limited, and further studies with larger cohorts are warranted to validate the findings. Future research should explore the long-term effects of Manuka honey on scar maturation and assess its efficacy in different types and severities of burns. Clinical trials involving human subjects are essential to confirm its translational potential and optimize application protocols.

Round 2

Reviewer 2 Report

Comments and Suggestions for Authors

The manuscript has been sufficiently improved according to my previous comments. The manuscript is qualified to publish.

Comments on the Quality of English Language

The English language of this manuscript is clear and coherent. 

Author Response

Dear Reviewer,

Thank you for your valuable feedback and for recognizing the improvements in our manuscript. We appreciate your positive assessment and are glad that you find the revised version suitable for publication. Your constructive insights throughout the review process have been truly helpful.

Best regards,
On behalf of the authors

Reviewer 3 Report

Comments and Suggestions for Authors

Thanks for the invitation to review this work. The authors have solved the previous concerns, and the article is recommended for publication after careful proof check.

Unify the capitalization of the first letters (Honey) and indent two spaces at the beginning of each paragraph.

Author Response

Dear Reviewer,

We appreciate your careful review and helpful suggestions. As per your recommendation, we have ensured consistency in capitalization (e.g., "Honey") and applied proper indentation to all paragraphs. Your attention to detail has helped refine the manuscript, and we are grateful for your input.

Best regards,
On behalf of the authors

Reviewer 4 Report

Comments and Suggestions for Authors

We thank the reviewers for addressing our comment and removing all references to scarring in this study. Now that the study is using this model to study wound healing, we suggest the following changes to improve this study.

1-       The authors should remove some of the pictures from Figure 5 to comply with ethical guidelines for animal use suggesting to not show images of whole animals. Specifically, the authors should remove Figures 5B, 5F, and 5G.

2-       Figure 4 should be modified to include individual dot plots to show biological variability between the animals and wounds for each treatment. The authors should also show the raw images and quantification data for all treatments.

3-       For Figure 3, The authors should show quantification for all the histology data for all treatment groups. This should be performed for Iba1, Ki67, and AE1/AE3.  The data should be presented in dot plots to show biological variability between the burn wounds and animals.

4-       The results state that: “The control group exhibited delayed epithelialization, higher inflammation, and less organized collagen structure, leading to poorer healing outcomes”. This is an overstatement, and the authors should present the data supporting these conclusions. Graphed data showing the quantification will help make the conclusions.

5-       The authors state that: “Histopathological evaluation showed superior dermal reconstruction with Manuka honey, increased collagen density, and horizontally oriented fibres, indicating better-quality tissue”. This is another overstatement as the authors did not show graphs and statistical analysis showing improvement with Manuka Honey. As mentioned above, please provide quantification of the images shown for H&E staining, Trichrome, and PRS. Specific freely available software can help with the quantification of inflammation, collagen density, collagen alignment, and tissue quality. Please provide dot plots for these quantifications to show biological variability between the different burn sites/treatments and animals. 

Comments on the Quality of English Language

Can be improved 

Author Response

Comment 1: The authors should remove some of the pictures from Figure 5 to comply with ethical guidelines for animal use suggesting to not show images of whole animals. Specifically, the authors should remove Figures 5B, 5F, and 5G.

Answer 1: We appreciate the reviewers’ suggestion. In accordance with ethical guidelines, we have removed Figures 5B, 5F, and 5G from Figure 5 to ensure compliance with recommended standards for animal use in research. Additionally, we have included images of collagen density measurements to provide a more comprehensive visualization of the histological analysis.

Comment 2: Figure 4 should be modified to include individual dot plots to show biological variability between the animals and wounds for each treatment. The authors should also show the raw images and quantification data for all treatments.

Answer 2: Thank you for the suggestion. We have modified Figure 4 to include individual dot plots, which illustrate biological variability among animals and wounds for each treatment group. Additionally, we have provided the raw images and quantification data for all treatments to enhance transparency and reproducibility.

Comment 3: For Figure 3, The authors should show quantification for all the histology data for all treatment groups. This should be performed for Iba1, Ki67, and AE1/AE3.  The data should be presented in dot plots to show biological variability between the burn wounds and animals.

Answer 3: We have addressed this request by adding quantification for all histology data, including Iba1, Ki67, and epithelialization, presented as scatter plots. These visualizations display biological variability between burn wounds and animals, as requested. The quantification of AE1/AE3 was determined through H&E staining, as epithelial thickness was measured. This information has been incorporated into the methodology and results sections. Additionally, we have included a new Figure 5 containing the relevant data.

Comment 4:   The results state that: “The control group exhibited delayed epithelialization, higher inflammation, and less organized collagen structure, leading to poorer healing outcomes”. This is an overstatement, and the authors should present the data supporting these conclusions. Graphed data showing the quantification will help make the conclusions.

Answer 4: We have revised the sentence to ensure it accurately reflects the quantified data. The updated wording removes any overstatements and is now fully aligned with the presented results. Additionally, we have modified the abstract (Results section) to clearly present the quantified findings in a more precise manner.

Comment 5:     The authors state that: “Histopathological evaluation showed superior dermal reconstruction with Manuka honey, increased collagen density, and horizontally oriented fibres, indicating better-quality tissue”. This is another overstatement as the authors did not show graphs and statistical analysis showing improvement with Manuka Honey. As mentioned above, please provide quantification of the images shown for H&E staining, Trichrome, and PRS. Specific freely available software can help with the quantification of inflammation, collagen density, collagen alignment, and tissue quality. Please provide dot plots for these quantifications to show biological variability between the different burn sites/treatments and animals. 

Answer 5: We have revised this statement to ensure that the conclusions accurately reflect the available quantitative data. Additionally, we have modified sentences in the Discussion section to remove any overstatements and ensure that all claims are fully supported by the presented results.

We have provided quantification of collagen density using histopathological analysis, with results now presented in scatter plots to illustrate biological variability between different burn sites, treatments, and animals. Inflammation was assessed using Iba1 immunohistochemistry, and its quantification has been included in the results.

However, collagen fiber orientation was not analyzed, as the available image data did not allow for consistent quantification. We have clarified this in the revised manuscript.

All modifications requested in the second round of review have been highlighted in orange within the revised manuscript for clarity.

Reviewer 5 Report

Comments and Suggestions for Authors

Dear Authors,

Thank you for your unwavering dedication and the effort you have put into improving your manuscript. My intention in highlighting certain aspects was not to create difficulties for you but to enhance the quality and clarity of your work. After reviewing your responses and the revisions made, I believe the manuscript has significantly improved, and I have no objection to recommending its publication. I wish you success with this work and in your future research endeavors.

Author Response

Dear Reviewer,

We are truly grateful for your thoughtful review and encouraging words. Your feedback has helped us enhance the manuscript, and we appreciate your support in recommending it for publication. Thank you for your time and constructive guidance.

Best regards,
On behalf of the authors